# Rapid purification of brain protein complexes containing active and inactive forms of the G protein Gα$_o$

Shubham Yadav [1,2], Satya Santoshi Veliventi[1], Sitaram Meena[3], Michael R. Koelle[4], Santosh Kumar [1,2]*

1 National Centre for Cell Science, Savitribai Phule Pune University, Maharashtra , India, 2 Regional Centre for Biotechnology, NCR Biotech Science Cluster, Faridabad, Haryana, India, 3 UCB Pharma, Cambridge, Massachusetts, United States of America, 4 Yale University, New Haven, Connecticut, United States of America

* sbalot@nccs.res.in, sbalot@gmail.com

## Abstract

Gα$_o$ is the alpha subunit of the most abundant heterotrimeric G protein of the brain and relays signals from G protein-coupled receptors to inhibit neural function. To date, no direct downstream effectors for Gα$_o$ have been well-characterized. Active Gα-GTP proteins should form stable complexes with their effectors, but identifying Gα$_o$ effectors by isolating such complexes is a challenge since the vast majority of Gα$_o$ in the brain is in its inactive, GDP-bound form. In this study, we developed methods to isolate microgram quantities of native Gα$_o$-GTP protein complexes by immunoprecipitation from brain lysates. We found that native Gα$_o$ protein in crude detergent lysates of mouse brain rapidly binds and is activated by the slowly-hydrolyzable GTP analog GTPγS. Using size exclusion chromatography and tracking the Gα$_o$-containing complexes by western blotting, we found that native Gα$_o$ in crude brain lysates exists in complexes that change size upon activation by GTPγS. We also identified a monoclonal antibody that can efficiently immunoprecipitate Gα$_o$ protein complexes from mouse brain lysates for downstream applications such as mass spectrometry and protein-protein interaction assays. Our results and methods enable further research into the Gα$_o$ signaling pathway.

G protein-coupled receptors (GPCRs) transduce extracellular stimuli into intracellular signals. In the brain, GPCRs control cognitive and behavioral functions by mediating the effects of neurotransmitters, neuropeptides, and other signaling molecules [1]. GPCR signaling is mediated by heterotrimeric G proteins consisting of three subunits: α, β, and γ. The downstream response elicited by a GPCR is primarily dictated by the type of G protein α (Gα) subunit activated. Gα subunits in animals and humans are classified into four subfamilies: Gα$_s$, Gα$_{q/11}$, Gα$_{12/13}$, and Gα$_{i/o}$ [2]. GPCRs activate

**Data availability statement:** All relevant data are within the manuscript and its Supporting Information files.

**Funding:** S.K.'s laboratory is funded by Department of Biotechnology (DBT), Ministry of Science and Technology India (BT/PR38584/MED/122/247/2020), Department of Science and Technology (DST), Ministry of Science and Technology India (CRG/2021/000732) and DBT/Wellcome Trust India Alliance (IA/I/22/2/506480). M.R.K. is supported by NIH grant RO1NS086932. The funders were not involved in the design of the study, the collection or interpretation of data, the publication decision, or the preparation of the manuscript.

**Competing interests:** The authors declare that there are no competing financial interests.

G proteins by catalyzing the release of guanosine diphosphate (GDP) from the Gα subunit, allowing it to instead bind guanosine triphosphate (GTP) and triggering the release of Gβγ [3]. Activated Gα-GTP proteins can then form complexes with downstream effector molecules that may function as enzymes, ion channels, or in other ways to regulate intracellular signaling pathways [4].

$Gα_o$ is by far the most abundant G protein found in brain tissue [5], and signals to inhibit neural function [6–8]. However, $Gα_o$ remains the only type of Gα protein in higher eukaryotes for which an effector has not been well characterized. Gβγ complexes released by activated $Gα_o$ can regulate ion channels and soluble N-ethylmaleimide-sensitive factor-activating protein receptor proteins (SNARE), a family of proteins critical for synaptic vesicle fusion and neurotransmitter release. Gβ can directly bind to SNARE proteins, leading to the inhibition of exocytosis after $Ca^{2+}$ influx in the cell [9–12]. Genetic studies in *Caenorhabditis elegans* (*C. elegans)* suggest that the major behavioral effects of $Gα_o$ signaling occur via regulation of effectors activated directly by the $Gα_o$ subunit itself, rather than via its release of Gβγ [4]. Several proteins including PINS (Partner of Inscuteable), Rap-1 GTPase activating protein (Rap1GAP), Regulator of G protein signaling (RGS17), and Ras-related protein in brain (Rab5)-interacting partners, have been proposed as $Gα_o$ effectors [13–16], but these, as yet, have not been established by genetic studies to mediate the primary effects of $Gα_o$ signaling on neural function. Some studies in yeast suggested that $Gα_o$ may be unique among G proteins and may not have an effector. Thus, $Gα_o$ signaling could occur solely via the release of Gβγ subunits upon activation of $Gα_o$, since free Gβγ is known to bind and regulate certain $Ca^{2+}$ and $K^+$ channels [17–18]. However, this idea is refuted by *C. elegans* genetics because *C. elegans* $Gα_o$, unlike the yeast G proteins, does not signal solely by releasing Gβγ and must rather signal via effector(s) activated by the $Gα_o$ subunit itself [6,7]. In addition, *C. elegans* has all four classes of Gα subunits seen in mammals, whereas yeast only has one. Mutations in each type of Gα have different effects. As in mammals, all worm Gα proteins associate with the same Gβγ dimers [4]. If $Gα_o$ signals by releasing Gβγ, it is unclear how it differs from other Gα proteins that also release the same Gβγ subunits.

Genetic studies in *C. elegans* have identified proteins that may act downstream of $Gα_o$ to mediate the signaling [19–21], but these have not been shown to be directly bound or regulated by $Gα_o$, as required if they are to meet the biochemical criteria for acting as $Gα_o$ effectors. Genetic screens might fail to identify $Gα_o$ effectors if $Gα_o$ has several functionally redundant effectors, or if mutations in effector genes are lethal. Alternatively, the phenotypes could often be masked by compensatory pathways. Several candidates identified by genetic screens have not been confirmed due to a lack of biochemical interaction data or rescue experiments. Interaction screens, such as the yeast two-hybrid system, have identified several proteins that appear to directly bind $Gα_o$ [22–24], but these interacting proteins have not yet been established as mediating the major effects of $Gα_o$ signaling in the brain. Such interaction screens might fail to identify $Gα_o$ effectors since most known G protein effectors are membrane protein complexes, which are unlikely to be detected in yeast two-hybrid screens. Moreover, the folding and post-translational modifications of mammalian $Gα_o$ may not be properly

replicated in yeast, potentially preventing correct interactions. These screens also tend to generate false positives and miss transient or low-affinity interactions, which are common in G protein signaling. To date, the only identified effectors of $G\alpha_o$ are GPRIN-1(G protein-regulated inducer of neurite outgrowth 1) and Necdin [25–26], discovered through studies in cell lines and mouse models. However, these effectors appear to be system-specific and are yet to be fully validated through biochemical approaches, leaving the search for a universally conserved downstream effector ongoing.

In this study, we developed methods for purifying native $G\alpha_o$-GTP/effector protein complexes from brain lysates, with the aim of overcoming the limitations of genetic and interaction screens discussed above. In contrast, our biochemical purification approach works directly with solubilized mouse brain lysates in near-physiological buffer conditions. This enables the native post-translational modifications, lipid environments, and protein complexes to remain intact, allowing detection of physiologically relevant complexes. By stabilizing $G\alpha_o$ in its active, GTPγS-bound form, we selectively enrich for complexes that include downstream effectors. It provides a powerful complement to genetic approaches and opens the door for mass spectrometry-based identification of bona fide $G\alpha_o$ effectors that may have been missed in prior screens.

## Materials and methods

Key reagents are listed in the Supporting Information, **S1 Table**. All the experiments were performed twice unless otherwise indicated.

### Ethics statement

All animal-related experiments were reviewed and approved by the Institutional Animal Ethics Committee (EAF/2021/B-429) and carried out in strict accordance with the guidelines set by the Committee for the Purpose of Control and Supervision of Experiments on Animals (CPCSEA), Government of India. Prior to the initiation of experimental procedures, mice were housed in individual ventilated cages (IVCs) under standard laboratory conditions, including a 12-hour light/dark cycle, regulated temperature and humidity, and were provided with sterilized food and water *ad libitum*. Each IVC was equipped with corn cob bedding and provided with environmental enrichment to promote animal welfare. During the experiment, the animals were carefully removed from the cage and restrained appropriately. For euthanasia, sodium pentobarbital was injected intraperitoneally in mice and subsequently monitored for the loss of righting reflex, followed by indicators of death. All procedures were conducted under sterile, controlled conditions by skilled and experienced personnel, with rigorous observance to minimize animal pain and distress throughout the study.

### Preparation of detergent-solubilized whole brain extracts

Mouse brain lysates were prepared as previously described [27]. Briefly, four BALB/c adult mice (21 days old) were euthanized and processed for lysate preparation, as described above. The cerebellum was removed, and the brains were sliced in half. Each brain was homogenized in 1.67 ml homogenization buffer [20mM HEPES pH 7.4, 25mM potassium acetate, 320 mM sucrose, 1% Triton X-100, and protease inhibitors (aprotinin, benzamidine, chymostatin, leupeptin, phenylmethanesulfonyl fluoride, and pepstatin A)], incubated on ice for 30 minutes, centrifugated at 100,000 g for 1 hour at 4°C, and the supernatant was collected. To determine protein concentrations, 10 μl of mouse brain lysate was diluted (1:10 and 1:20) in homogenisation buffer (without Triton X-100 and protease inhibitors) and measured using the Bradford Assay (Bio-Rad, USA). Each experiment was conducted using four mouse brains, and the entire experiment was repeated independently three times. We obtained an average concentration of ~12 mg/ml. The whole brain extract was aliquoted and snap-frozen. Prior to use, aliquots were thawed and diluted to 2 mg/ml total protein in the immunoprecipitation (IP) buffer.

### *In-vitro* activation and Immunoprecipitation (IP) of $G\alpha_o$

For immunoprecipitation, binding of a mouse monoclonal anti-$G\alpha_o$ antibody to protein A/G Agarose, followed by disuccinimidyl suberate (DSS) crosslinking, was performed using Pierce crosslink immunoprecipitation kits (Thermo Scientific)

per manufacturer's instructions with some modifications. Briefly, for each sample, 30 µl protein A/G agarose slurry was washed three times with 1x Phosphate Buffered Saline (PBS) buffer and then incubated with 10 µg mouse monoclonal anti-$G\alpha_o$ antibody sc-13532 (Santa Cruz Biotechnology, USA) in PBS buffer at room temperature (RT) for 2 hours on a rotator for mixing. Beads were washed three times with PBS buffer, followed by incubation with DSS working solution (50 µl) at room temperature (RT) for 1 hour on a rotator to stably crosslink antibodies to the agarose beads. The supernatant was discarded, and the beads were washed twice with pH 2.8 elution buffer (from the Pierce kit) and then twice with IP buffer (50mM HEPES pH 7.4, 100mM NaCl, 1mM EDTA, 3mM EGTA, 10mM $MgCl_2$ and 1% Triton X-100). In parallel, whole mouse brain extract (diluted to 2 mg/ml in IP buffer, 1 ml per IP) was incubated with protein A/G beads (15 µl packed beads for each 1 ml of diluted brain extract) to pre-clear the lysate, discarding the beads. The antibody cross-linked beads were then mixed with pre-cleared lysate, and either GDP (Sigma, USA) or GTPγS (BIOLOG, Life Science Institute, or Abcam) was added from 10 mM stock solutions to a final concentration of 100 µM. The tubes were incubated at 4 °C for 2 hours on a rotator to allow the binding of GDP or GTP nucleotide to $G\alpha_o$ and proceeded simultaneously with binding of $G\alpha_o$ to the antibody on the beads. After removing the supernatant, the beads carrying $G\alpha_o$ protein complexes were washed four times with IP buffer and eluted with 2X Lithium dodecyl sulfate (LDS) loading buffer (Invitrogen) lacking β-mercaptoethanol (50 µl) at 55 °C for 30 min. The supernatants were transferred into the new tubes, 2.5% β-mercaptoethanol was added, and the tubes were further incubated at 55 °C for 15 min. These IP samples were then analyzed by SDS-PAGE and western blotting.

### SDS-PAGE and western blotting

For SDS-PAGE, 60% of each IP sample was loaded on a lane of a NuPAGE 4–12% Bis-Tris's gel (Invitrogen) and separated using NuPAGE MOPS SDS buffer (Invitrogen). Gels were stained with Imperial Protein Stain (Thermo Scientific, USA) as per the manufacturer's instructions, and images were captured using an Epson Perfection V800 Photo Color Scanner at 800 dpi. Images were processed by ImageJ software. For western blots, 20% of each IP sample was loaded into a well and separated on a 10% SDS-PAGE gel. The protein bands were transferred to a nitrocellulose membrane, and blots were probed with 1:200 diluted mouse monoclonal anti-Gβ (Santa Cruz Biotechnology, USA), 1:5000 diluted sheep anti-GPRIN1 antibody (Kindly provided by Dr. Elhadji M Dioum, Nestlé Institute of Health Sciences, Switzerland), and 1:1000 diluted polyclonal rabbit anti-$G\alpha_o$ antibodies [28]. Proteins were visualized with enhanced chemiluminescence horseradish peroxidase–linked secondary 1:5000 diluted anti-mouse (Bio-Rad, USA), 1:5000 diluted anti-sheep (Thermo Scientific, USA), and 1:5000 diluted anti-rabbit antibodies (Bio-Rad, USA) and Super Signal West Pico/Femto Maximum Sensitivity substrate (Thermo Scientific, USA).

### Size exclusion chromatography

$G\alpha_o$ protein was activated in mouse brain extracts by incubation with 100 µM GTPγS at 4° C for 2 hrs. As a negative control, mouse brain lysates were instead incubated with 100 µM GDP. After this incubation, samples were passed through a 0.2 µm filter, centrifuged at 70000g for 20 min at 4°C and loaded on the Superose™ 6 10/300 GL column (Cytiva, USA), which was preequilibrated with running buffer (50 mM HEPES, pH 7.4, 150 mM NaCl, 1 mM EDTA, 3 mM EGTA, 1 mM DTT, 10 mM $MgCl_2$ 0.1% Triton X-100, 1µM GDP or GTPγS). The flow rate was maintained at 0.5 ml/min and 0.5 ml fractions were collected per tube. Bed and void volumes of the column were 24 ml and 7 ml, respectively. After the void volume (fractions 1–14), all the fractions were collected. Samples were processed for western blotting with mouse monoclonal anti-Gβ and rabbit anti-$G\alpha_o$ antibodies as mentioned above.

### Mass spectrometry and data analysis

The immunoprecipitated samples were processed for mass spectrometry analysis by Yale Keck MS & Proteomics Resource. Eluted IP proteins were loaded on 4–12% Bis gel and run samples for 2 min at 200 volts. Bands were cut from

gel, and gel slices were cut into small pieces and washed on a tilt-table with 1 ml water for 10 minutes. The gel pieces were then washed for 30 min with 500 µl 50% acetonitrile (ACN)/100 mM $NH_4HCO_3$, then washed twice more for 15 minutes with 500 µl 50% ACN/25 mM $NH_4HCO_3$ The gels were briefly dried by SpeedVac, then resuspended in 180 µl of 25mM $NH_4HCO_3$ containing 350 ng of digestion-grade trypsin (Promega, V5111) and incubated at 37ºC for 16 hours. The supernatant containing tryptic peptides was transferred to a new Eppendorf tube, and the gel band was extracted with 300 µl of 80% acetonitrile/0.1% trifluoroacetic acid (TFA) for 15 minutes. Supernatants were combined and dried by speed vacuum. Peptides were dissolved in 40 µl MS loading buffer (2% ACN, 0.2% TFA), with 5 µl injected for LC-MS/MS analysis. Tandem mass spectra were extracted by Proteome Discoverer software (version 2.2.0.388, Thermo Scientific) and searched in-house using the Mascot algorithm (version 2.6.1, Matrix Science). The mouse data were searched against the SwissProt database (version 2018_04) with taxonomy restricted to *Mus musculus* (16,977 sequences). Scaffold Q+S (version 4.8.9, Proteome Software Inc., Portland, OR) was used to validate MS/MS-based peptide and protein identifications.

## Results and discussion

### Immunopurification of Gα$_o$ from mouse brain lysate after activation with GTPγS

We prepared mouse brain lysates using whole adult brains by gentle homogenization in buffer containing a non-ionic detergent to solubilize Gα$_o$ and other membrane proteins with which it might associate (Fig 1A). More than 90% of Gα$_o$ in the brain is in its inactive, GDP-bound form [29]. While purified Gα proteins can dissociate from GDP and then bind to GTP or its slowly-hydrolysable analog, GTPγS, typically, an extended incubation in buffer containing very high $Mg^{2+}$ concentrations (>25 mM) and at a temperature of >32° is required to induce a significant level of this nucleotide exchange. However, nucleotide exchange by Gα$_o$ is much faster and less $Mg^{2+}$ dependent [14]. We determined, as shown in Fig 1, that treating brain extract with 100 µM GTPγS in a buffer containing a more physiological $Mg^{2+}$ concentration (5 mM) and incubating at 4 °C for 2 hours was sufficient to induce nearly 100% activation of GTPγS in the extract.

To rapidly and efficiently enrich Gα$_o$ protein complexes from the total brain extract for analysis, we immunoprecipitated Gα$_o$ with a commercially available mouse anti-Gα$_o$ antibody (sc-13532) that proved highly efficient for this purpose. When the anti-Gα$_o$ monoclonal antibody was omitted from the immunoprecipitation (IP) protocol, no detectable proteins were pulled down in this control, as seen in a total protein stain after separation of proteins by SDS-PAGE sample (Fig 1B, control lane). Using the anti-Gα$_o$ antibody, from 2 mg of total brain protein preincubated with GDP, immunoprecipitation yielded several micrograms of Gα$_o$ protein at a level of purity that allowed the Gα$_o$ (40 kDa), Gβ (37 kDa), and Gγ (subunits to be easily visualized, with Gα$_o$ being the most abundant single protein seen (Fig 1B, lane "GDP"). The stain intensities of these bands on the gel are consistent with Gα$_o$, Gβ, and Gγ being present in a complex at 1:1:1 stoichiometry, as expected for the inactive GDP-bound G protein. When the brain lysate was preincubated with GTPγS rather than GDP prior to IP, a similar amount of Gα$_o$ protein was pulled down, but no Gβ or Gγ was detectable in the immunoprecipitate (Fig 1B, lane "GTPγS"). These results suggest that the incubation with GTPγS under the mild conditions we used was sufficient to activate virtually all of the Gα$_o$ in the extract, since active Gα$_o$-GTP is expected to dissociate from Gβγ.

We confirmed the identification of the Gα$_o$ and Gβ bands in the protein stain and the activation state of Gα$_o$ using western blotting. Probing with an anti-Gα$_o$ polyclonal antibody, we confirmed that the major band at 40 kDa was indeed Gα$_o$ and that it was present at roughly equal levels in both the GDP and GTPγS pull-down samples, while absent in the no-antibody control pull-down sample (Fig 1C, lower panel). Probing the same blot with an anti-Gβ subunit antibody, we detected the Gβ subunit as the major 37 kDa band in the GDP pull-down sample and saw that it was absent in the GTPγS pull-down sample (Fig 1C, upper panel), confirming that incubation of the brain extract with GTPγS under mild buffer conditions at 4° C caused essentially complete activation of the Gα$_o$ protein in the extract. Quantification (Fig 1D) further confirms a substantial decrease in Gβ binding in the GTPγS condition, normalized to the amount of Gα$_0$ pulled down, validating that the observed difference is due to nucleotide-dependent complex dissociation.

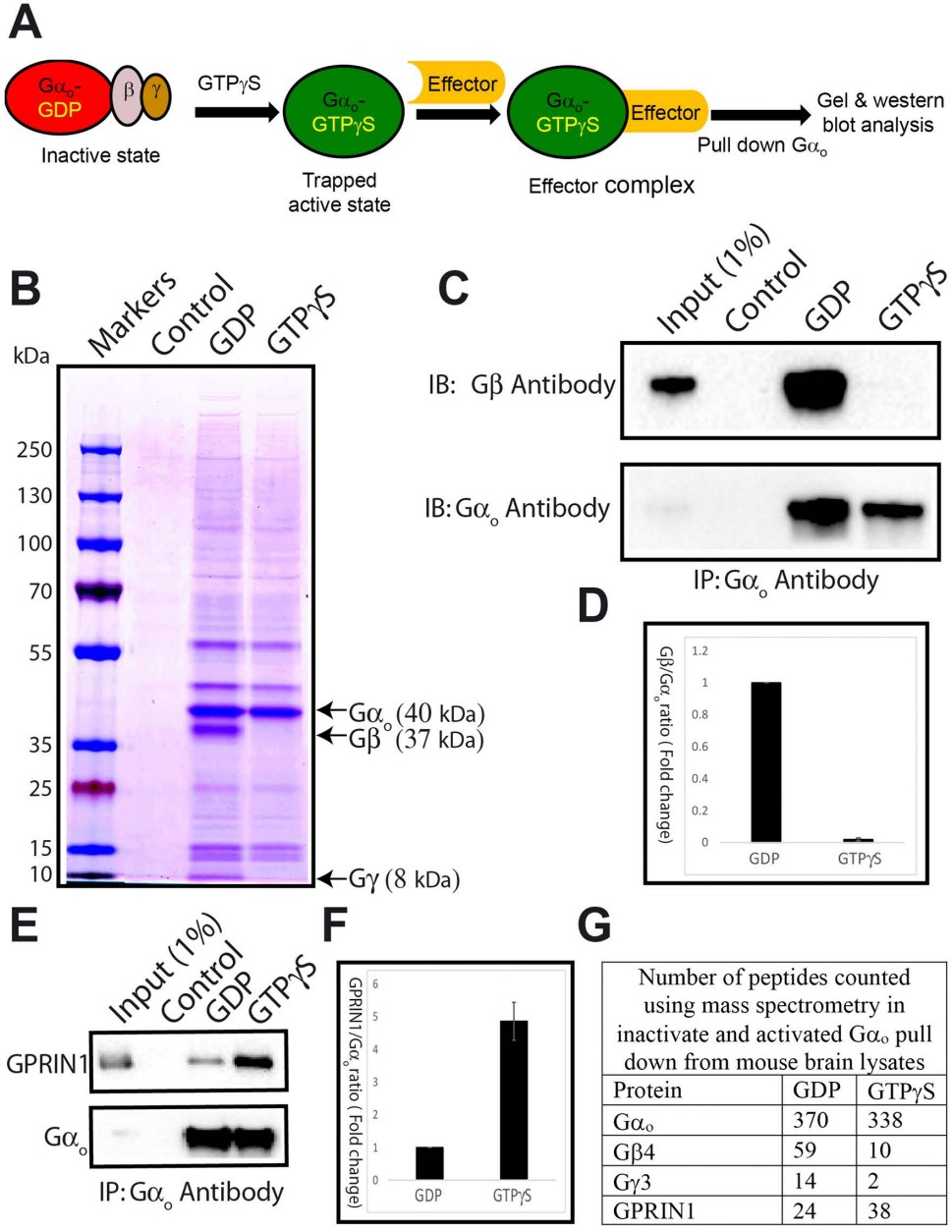

**Fig 1. Rapid Immunopurification of active and inactive Gα$_o$ protein complexes from brain lysates. (A)** Schematic representing activation and immunoprecipitation of Gα$_o$ from whole adult mouse brain extracts. **(B-C)** Mouse brain lysates were prepared and processed for immunoprecipitation using an anti-Gα$_o$ antibody as described in the Materials and Methods section. **(B)** Immunoprecipitated samples were analyzed by SDS-PAGE, and Imperial protein stain was used to visualize protein bands separated by the gel. Gα$_o$, Gβ, and Gγ bands are indicated. Other bands in the gel are either background contaminants or proteins that bind to Gα$_o$. **(C)** After immunoprecipitation with an anti-Gα$_o$ monoclonal antibody, immunoprecipitated samples were resolved by SDS-PAGE, transferred to nitrocellulose, and probed with a mouse monoclonal anti-Gβ antibody and rabbit polyclonal anti-Gα$_o$ antibody on western blots.. **(D)** The mean intensity of the Gβ and Gα$_o$ protein bands shown in panel C was measured using ImageJ software. The mean intensity of the Gβ protein band was normalized against Gα$_o$. The data is represented as a ratio of Gβ to Gα$_o$ (Mean intensity ± SD). **(E)** Samples for the western blot were prepared as described in panel C and probed with sheep anti-GPRIN1 antibody and rabbit polyclonal anti-Gα$_o$ antibody. **(F)** The mean intensity of the GPRIN1 protein band was quantified, and the data were normalized with Gα$_o$ as mentioned in panel **D**. **(G)** Following immunoprecipitation with an anti-Gα$_o$ monoclonal antibody, the samples were prepared for mass spectrometry as detailed in the Materials and Methods section. The number of peptides in both inactive and activated Gα$_o$ pull-down samples was measured using Scaffold (TM) version: Scaffold_5.0.1, with a protein and peptide threshold set to 95%.

If the $Ga_o$-GTPγS was stably bound in a stoichiometric complex with a single type of effector protein, we would have expected to see a Coomassie-stained band for that effector protein present in the GTPγS lane in the gel shown in Fig 1B. Such a band would be present at the same abundance as the $Ga_o$ band, but absent from the GDP lane. We did not see such a band, but given the extremely high abundance of $Ga_o$ in the brain, it is unlikely that a $Ga_o$ effector could be present at anywhere near the same abundance as $Ga_o$. Thus, a small portion of the $Ga_o$-GTPγS in the brain lysate may indeed be bound to effector(s), such that these lower abundance effectors are not detectable via Coomassie staining, but may be detectable by more sensitive methods such as mass spectrometry.

To explore the presence of such effectors, we examined GPRIN-1, a previously identified candidate $Ga_o$ effector [25], by western blotting after IP of $Ga_o$. As shown in Fig 1E, GPRIN-1 was detected robustly in the GTPγS condition but not in the GDP condition or control lane. This selective enrichment indicates that GPRIN-1 preferentially associates with the active, GTPγS-bound form of $Ga_o$. Quantification (Fig 1F) showed a marked increase in GPRIN-1 signal in the GTPγS-bound sample normalized to $Ga_o$ input, further supporting this nucleotide-dependent interaction.

To comprehensively evaluate the composition of $Ga_o$ protein complexes under GDP and GTPγS conditions, we performed mass spectrometry on the immunoprecipitated samples. As summarized in Fig 1G, mass spectrometry analysis confirmed efficient pull-down of $Ga_o$ in both conditions, as evidenced by the high peptide counts (370 and 338 peptides in GDP and GTPγS samples, respectively). Gβ and Gγ were readily detected in the GDP-bound complex (59 and 14 peptides, respectively), but peptide counts dropped sharply in the GTPγS condition (10 and 2 peptides, respectively), confirming dissociation of the heterotrimer. Importantly, the number of peptides corresponding to GPRIN-1 increased in the GTPγS sample (38 vs. 24 in GDP), consistent with its preferential binding to the activated form of $Ga_o$.

The selective interaction of GPRIN-1 with the activated form of $Ga_o$ (Fig 1E and Fig 1G) suggests it plays a functional role downstream of $Ga_o$ signaling, potentially mediating effects on neuronal growth or synaptic plasticity, as previously hypothesized.

## Analysis of $Ga_o$-GTPγS protein complexes by size exclusion chromatography

While immunoprecipitation and mass spectrometry provided valuable insights into nucleotide-dependent composition of the $Ga_o$ complexes, they did not reveal information about the size or overall organization of those complexes. To address this, we next examined the $Ga_o$-GDP and $Ga_o$-GTPγS complexes using size exclusion chromatography, allowing us to assess potential changes in complex size and assembly state upon activation. We preincubated $Ga_o$ in detergent-solubilized mouse brain lysates with GTPγS or GDP, as in Fig 1, and then used size-exclusion chromatography to analyze the apparent sizes of $Ga_o$ protein complexes (Fig 2A). Looking at the profile of total protein eluting from the column, the pattern was similar whether we had pretreated with GDP or GTPγS (Fig 2B). To identify the eluted fractions containing inactive $Ga_o$-GDP/Gβγ and active $Ga_o$-GTPγS, we analyzed each fraction using western blotting with $Ga_o$ and Gβ subunits antibodies.

In the presence of GDP, a condition in which the IP analysis of Fig 1 shows that $Ga_o$-GDP and Gβγ are in a complex, we saw that $Ga_o$ and Gβ comigrated, as expected (Fig 2C, lower blots). However, they were mostly present in fractions 15–20, corresponding to an apparent molecular weight of ~1249–945 kDa. This is much larger than expected based on the molecular weight of the $Ga_o$/Gβγ heterotrimer (~90 kDa), even taking into account that the G protein complex is expected to be incorporated into micelles of the Triton X-100 detergent included in the buffer. After activation with GTPγS, we subjected the samples to high-speed ultra-centrifugation before loading on the column; however, we still acknowledge the possibility that a portion of the $Ga_o$ complexes may be present in higher-order oligomers or transient aggregates, which could further change the apparent molecular weights. In the presence of GTPγS (Fig 2C, upper blots), the majority of $Ga_o$ protein shifted to lower molecular weight fractions (29–34), corresponding to apparent molecular weights in the ~98–398 kDa range, again larger than the ~45 kDa expected for a $Ga_o$ monomer, although the presence detergent micelles must again be taken into account. A small amount of the Gβ subunit was present in these fractions (Fig 2C upper

# A

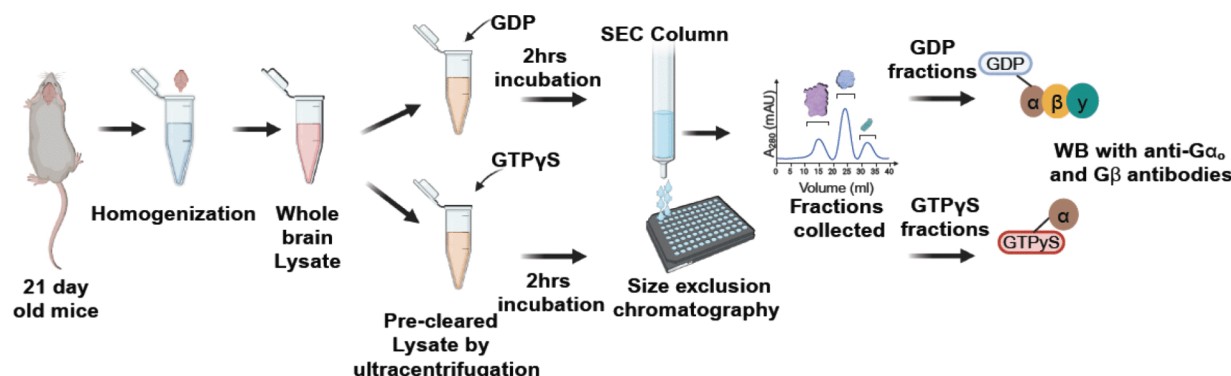

# B

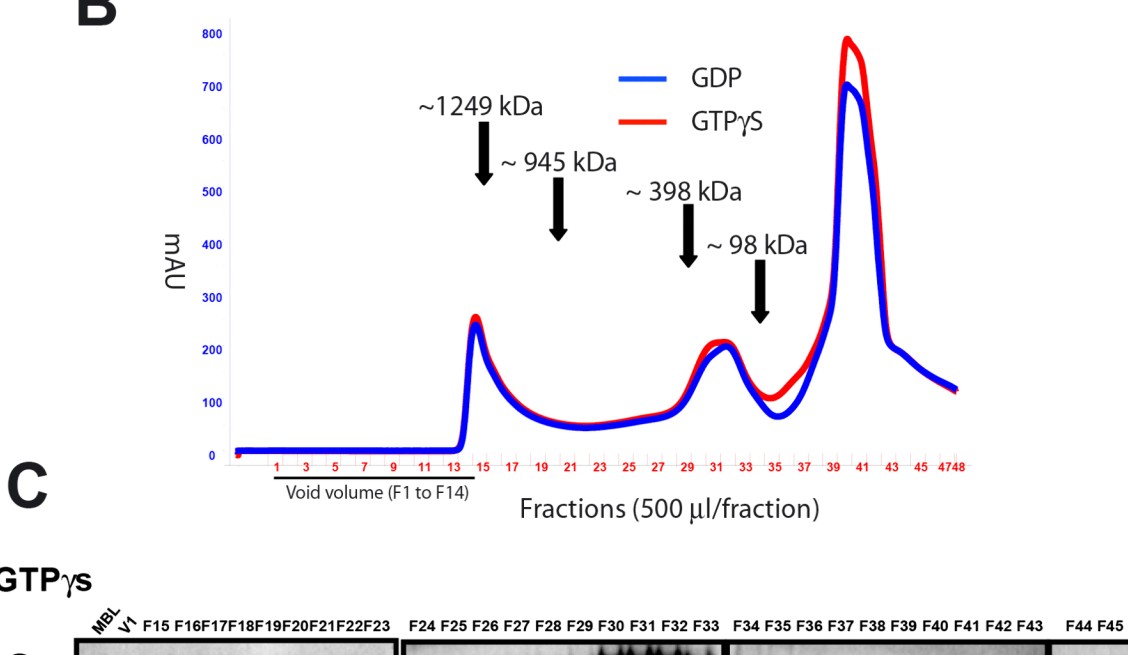

# C

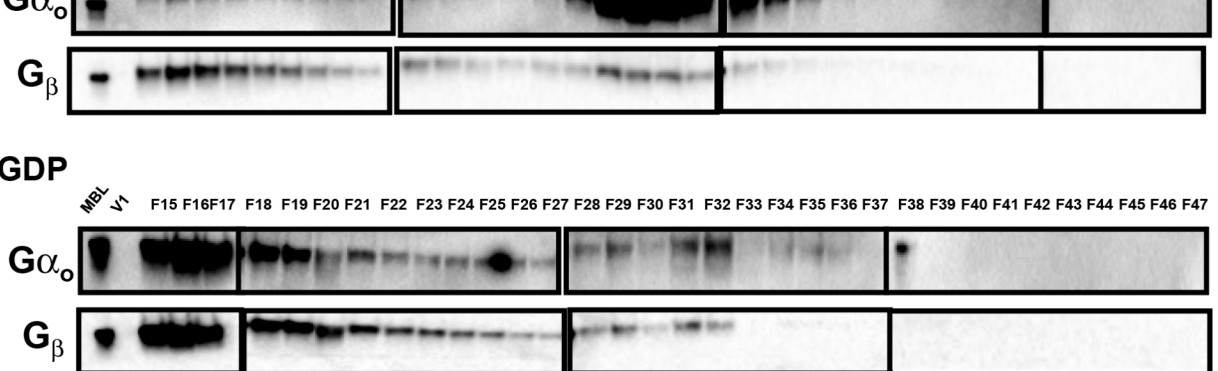

**Fig 2. Analysis of native mouse brain protein complexes containing active or inactive forms of Gα$_o$ by size-exclusion chromatography. (A)** Schematic representing the experimental workflow. Whole brain lysates from 21-day-old mice were homogenized and pre-cleared by ultracentrifugation.

They were further incubated with 100 µM GDP as a control or with 100 µM GTPγS to activate Gα$_o$. Following activation, lysates were fractionated using size-exclusion chromatography. Fractions were collected and analyzed by immunoblotting using anti-Gα$_o$ and anti-Gβ antibodies. **(B)** SEC Chromatogram showing UV absorbance (280nm) profiles of brain lysates incubated with GDP (blue) and GTPγS (red). The elution fractions of molecular weight standards are indicated. (C) 2% of the fractionated samples were separated on 10% SDS-PAGE and then processed for western blotting. Blots were probed with mouse monoclonal anti-Gα$_o$ and anti-Gβ antibodies. In the active state, Gα$_o$ was detected in the later fractions (F29 to F34), while in the inactive state, Gα$_o$ protein was detected in much earlier fractions (F15–F20). A 1% mouse brain lysate (MBL) input sample was used, and V1 denotes the first fraction of void volume.

blots), although the IP results of Fig 1 suggest it is not tightly bound to Gα$_o$-GTPγS. The size of the Gα$_o$-GTPγS complexes suggests that after activation by GTPγS, Gα$_o$ is found in complex with binding partners, which could include the unknown effector molecules for Gα$_o$ signaling. Thus, these protocols for *in vitro* activation and immunoprecipitation of Gα$_o$ from brain lysates can be used for downstream applications such as mass spectrometry to identify Gα$_o$ binding partners.

## Supporting information

**S1 Table. Key reagents used in this study.**
(DOCX)

**S1 File. Original blots and raw gel images.**
(PDF)

**S1 Figure. Graphical abstract.**
(PDF)

## Acknowledgments

We thank Dr Prasad Abnave (NCCS, Pune) for critical evaluation and valuable feedback on the manuscript. We heartily thank our lab members, Ajay Pradhan, Vandna Maurya, and Debolina Sarkar for their continuous and generous support, and Nikhat Khan for skillful technical assistance. We also thank Dr. Rahul Bankar (NCCS, Pune) for the technical assistance related to mouse handling.

## Author contributions

**Conceptualization:** Michael R. Koelle, Santosh Kumar.

**Funding acquisition:** Michael R. Koelle, Santosh Kumar.

**Investigation:** Shubham Yadav, Satya Santoshi Veliventi, Sitaram Meena, Santosh Kumar.

**Methodology:** Santosh Kumar.

**Project administration:** Michael R. Koelle, Santosh Kumar.

**Supervision:** Michael R. Koelle, Santosh Kumar.

**Writing – original draft:** Santosh Kumar.

**Writing – review & editing:** Michael R. Koelle.

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
