## [Decision Letter · Decision Letter 0]

19 Jun 2025

Dear Dr. Kumar,

Thank you for submitting your manuscript to PLOS ONE. After careful consideration, we feel that it has merit but does not fully meet PLOS ONE’s publication criteria as it currently stands. Therefore, we invite you to submit a revised version of the manuscript that addresses the points raised during the review process.

We look forward to receiving your revised manuscript.

Kind regards,

Ajit Prakash, PhD

Academic Editor

PLOS ONE

**Journal Requirements:**

1. When submitting your revision, we need you to address these additional requirements. Please ensure that your manuscript meets PLOS ONE's style requirements, including those for file naming. The PLOS ONE style templates can be found at https://journals.plos.org/plosone/s/file?id=wjVg/PLOSOne_formatting_sample_main_body.pdf and https://journals.plos.org/plosone/s/file?id=ba62/PLOSOne_formatting_sample_title_authors_affiliations.pdf 2. To comply with PLOS ONE submissions requirements, in your Methods section, please provide additional information regarding the experiments involving animals and ensure you have included details on (a) methods of sacrifice, (b) methods of anesthesia and/or analgesia, and (c) efforts to alleviate suffering. 3. Thank you for stating in your Funding Statement: S.K.’s laboratory is funded by Department of Biotechnology (DBT), Ministry of Science and Technology India (BT/PR38584/MED/122/247/2020), Department of Science and Technology (DST), Ministry of Science and Technology India (CRG/2021/000732) and DBT/Wellcome Trust India Alliance (IA/I/22/2/506480).  M.R.K. is supported by NIH grant RO1NS086932.  Please provide an amended statement that declares *all* the funding or sources of support (whether external or internal to your organization) received during this study, as detailed online in our guide for authors at http://journals.plos.org/plosone/s/submit-now.  Please also include the statement “There was no additional external funding received for this study.” in your updated Funding Statement. Please include your amended Funding Statement within your cover letter. We will change the online submission form on your behalf. 4. Thank you for stating the following in the Acknowledgments Section of your manuscript: We thank Dr Prasad Abnave (NCCS, Pune) for critical evaluation and valuable feedback on the manuscript. We heartily thank our lab members Ajay Pradhan, Vandna Maurya, and Debolina Sarkar for their continuous and generous support, and Nikhat Khan for skillful technical assistance. S.K.’s laboratory is funded by Department of Biotechnology (DBT), Ministry of Science and Technology India (BT/PR38584/MED/122/247/2020), Department of Science and Technology (DST), Ministry of Science and Technology India (CRG/2021/000732) and DBT/Wellcome Trust India Alliance (IA/I/22/2/506480).  M.R.K. is supported by NIH grant RO1NS086932. We note that you have provided funding information that is not currently declared in your Funding Statement. However, funding information should not appear in the Acknowledgments section or other areas of your manuscript. We will only publish funding information present in the Funding Statement section of the online submission form. Please remove any funding-related text from the manuscript and let us know how you would like to update your Funding Statement. Currently, your Funding Statement reads as follows: S.K.’s laboratory is funded by Department of Biotechnology (DBT), Ministry of Science and Technology India (BT/PR38584/MED/122/247/2020), Department of Science and Technology (DST), Ministry of Science and Technology India (CRG/2021/000732) and DBT/Wellcome Trust India Alliance (IA/I/22/2/506480).  M.R.K. is supported by NIH grant RO1NS086932.  Please include your amended statements within your cover letter; we will change the online submission form on your behalf. 5. PLOS requires an ORCID iD for the corresponding author in Editorial Manager on papers submitted after December 6th, 2016. Please ensure that you have an ORCID iD and that it is validated in Editorial Manager. To do this, go to ‘Update my Information’ (in the upper left-hand corner of the main menu), and click on the Fetch/Validate link next to the ORCID field. This will take you to the ORCID site and allow you to create a new iD or authenticate a pre-existing iD in Editorial Manager. 6. Please include your full ethics statement in the ‘Methods’ section of your manuscript file. In your statement, please include the full name of the IRB or ethics committee who approved or waived your study, as well as whether or not you obtained informed written or verbal consent. If consent was waived for your study, please include this information in your statement as well. 7. We notice that your supplementary table is included in the manuscript file. Please remove and upload with the file type 'Supporting Information'. Please ensure that each Supporting Information file has a legend listed in the manuscript after the references list. 8. Please include captions for your Supporting Information files at the end of your manuscript, and update any in-text citations to match accordingly. Please see our Supporting Information guidelines for more information: http://journals.plos.org/plosone/s/supporting-information. 9. PLOS ONE now requires that authors provide the original uncropped and unadjusted images underlying all blot or gel results reported in a submission’s figures or Supporting Information files. This policy and the journal’s other requirements for blot/gel reporting and figure preparation are described in detail at https://journals.plos.org/plosone/s/figures#loc-blot-and-gel-reporting-requirements and https://journals.plos.org/plosone/s/figures#loc-preparing-figures-from-image-files. When you submit your revised manuscript, please ensure that your figures adhere fully to these guidelines and provide the original underlying images for all blot or gel data reported in your submission. See the following link for instructions on providing the original image data: https://journals.plos.org/plosone/s/figures#loc-original-images-for-blots-and-gels.   In your cover letter, please note whether your blot/gel image data are in Supporting Information or posted at a public data repository, provide the repository URL if relevant, and provide specific details as to which raw blot/gel images, if any, are not available. Email us at plosone@plos.org if you have any questions.

Reviewers' comments:

Reviewer's Responses to Questions

**Comments to the Author**

1. Is the manuscript technically sound, and do the data support the conclusions?

Reviewer #1: Yes

Reviewer #2: Partly

2. Has the statistical analysis been performed appropriately and rigorously?

Reviewer #1: No

Reviewer #2: N/A

3. Have the authors made all data underlying the findings in their manuscript fully available?

Reviewer #1: No

Reviewer #2: Yes

4. Is the manuscript presented in an intelligible fashion and written in standard English?

Reviewer #1: Yes

Reviewer #2: Yes

**Reviewer #1: ** The manuscript authored by Yadav et al.., entitled “Rapid purification of brain protein complexes containing active and inactive forms of the G protein Gαo” presents a study aimed at identifying downstream effectors of the Gαo protein, the most abundant heterotrimeric G protein in the brain, by developing methods to isolate native Gαo-GTP protein complexes from mouse brain lysates. The authors employ immunoprecipitation and size exclusion chromatography to analyze Gαo complexes, demonstrating changes in complex size upon activation with GTPγS. The study addresses a significant gap in understanding Gαo signaling, as no direct effectors have been well-characterized despite its abundance and role in inhibiting neural function. The methods are novel and well-executed, but several areas require clarification, additional controls, and improved discussion to strengthen the manuscript's impact and clarity.

My comments are as follows:

1. The introduction mentions genetic and interaction screens that failed to identify Gαo effectors, but does not compare the current method’s advantages over these approaches in detail. A stronger rationale for why biochemical purification will likely succeed where genetic screens failed would strengthen the manuscript. Expand the introduction to discuss the limitations of yeast two-hybrid and genetic screens more explicitly and highlight how the current method overcomes these.

2. The manuscript aims to identify Gαo effectors but does not report any specific effector proteins identified through the described methods. The discussion mentions that effectors may be present at lower abundance and detectable by mass spectrometry, but no such data are provided. This is a significant limitation, as the study's primary goal appears unfulfilled. If mass spectrometry was performed, include preliminary results or clarify why such data are not presented. If not performed, explicitly state this as a limitation and outline plans for future work to identify effectors using these methods.

3. The observed molecular weights of Gαo complexes (1249–945 kDa for GDP-bound and 98–398 kDa for GTPγS-bound) are much larger than expected, even accounting for Triton X-100 micelles. The manuscript attributes this to potential effector interactions but does not adequately discuss alternative explanations, such as non-specific protein aggregation or micelle size variability. Include controls to rule out non-specific interactions, such as comparing Gαo complexes to those of other G proteins (e.g., Gαq or Gαi) under identical conditions. Discuss the impact of detergent micelles on apparent molecular weight more thoroughly, potentially referencing studies on membrane protein complexes in similar systems.

4. Validate anti-Gαo antibody specificity by testing it against lysates from Ga0 knockout mice or performing co-immunoprecipitation with other G protein subunits (e.g., Gαq or Gαi) to confirm selectivity.

5. The manuscript does not mention the number of biological or technical replicates for key experiments (e.g., immunoprecipitation, size exclusion chromatography). The statement “we prepared brain lysates multiple times” is vague. Specify the number of replicates and include statistical analysis where applicable (e.g., for protein yield or band intensity in western blots). Provide error bars or variability measures for quantitative data.

6. Figures 1B and 1C are referenced extensively, but the manuscript does not clearly describe the expected molecular weights of Gβ and G bands. The text mentions Gβ at 37 kDa and Ga0 at 40 kDa, but G is not quantified. Annotate figures with molecular weight markers and clarify G band identification in the text.

7. The term “GTPyS” is used in the abstract and early sections, while “GTP S” is used later. This inconsistency may confuse readers. Standardize to “GTP S” throughout the manuscript.

8. The manuscript presents a technically sound and novel approach to studying Gαo signaling by isolating native Gαo-GTP complexes from mouse brain lysates. The methods are well-executed and have the potential to advance the field, particularly if paired with downstream analyses like mass spectrometry. However, the lack of effector identification, incomplete validation of antibody specificity, and limited discussion of negative results weaken the current version. Addressing the major concerns (particularly providing effector data or clarifying its absence) and incorporating the minor suggestions will significantly strengthen the manuscript. I recommend major revisions before publication, focusing on clarifying results, adding controls, and enhancing the discussion.

**Reviewer #2:**  The authors have done a beautiful job of purifying complicated G�o protein from mouse brain lysate and tried to identify its binding to effectors in the manuscript. They have given a detailed introduction to the protein of interest and a detailed methodology of their experiments. Though the study in the manuscript is incomplete and inconclusive, it forms a base for future studies with these proteins.

I found the manuscript interesting, and it also projects interesting work. Having said that, there are some concerns about the manuscript, which would enhance the manuscript and help in future research directions even better. Below are my comments on the manuscript:

1. In the introduction part, it would help to provide some information about the effectors of G�o proteins. The authors have abruptly jumped into discussing effector proteins.

2. The authors have mentioned SNARE proteins. Nowhere in the manuscript have they described the abbreviation and its significance.

3. The authors say they prepared brain lysates multiple times to get 12 mg/ml. The scientific way of saying this would be to quantitate it by saying how many brain samples.

4. There are a couple of abbreviations for IP. The authors should either use the full form for one of them or should name the buffer with some other acronym.

5. The manuscript in general has errors/typos that need to be corrected. Also, the units should be written in a consistent manner throughout the manuscript like �L, mM, �M etc.

6. For SEC, the authors have mentioned the specific fraction groups collected but never mentioned the rationale of it. The better way of saying this would be all the fractions were collected.

7. The authors have mentioned GTP�S as a non-hydrolysable analog of GTP. However, it is a slowly hydrolyzing analog. For this study, better suited would have been other non-hydrolysable analogs such as GMPPCP/GMPPNP.

8. An alternative method to check the loading could have been HPLC, which would be more quantitative and unbiased.

9. In the western blot of fig. 1, the loading controls are absent, which is standard for the blot images.

10. In the blot of fig 1C, the G�o bands should be identical, as per the hypothesis and explanations in the text. Nothing has been discussed why do you see a difference in intensity of G�o in the GDP and GTP�S blots.

11. The author say mass spectrometry could be used to check the effector-binding. A lane in the SDS-PAGE with crude extract, which is not modified to load any nucleotide, would be an interesting sample to compare.

12. In fig.2, it is interesting and confusing at the same time, as to how there is a huge fraction from 37-45 which absorbs at 280, but does not read in Coomassie. It would be of interest to get the MALDI done for these samples and check the possibility. That may also give a conclusion to the manuscript.

**Do you want your identity to be public for this peer review?** For information about this choice, including consent withdrawal, please see our Privacy Policy

Reviewer #1: No

Reviewer #2: No

---

## [Author Response · Author response to Decision Letter 1]

3 Aug 2025

Manuscript ID: PONE-D-25-19360

Manuscript Title: Rapid purification of brain protein complexes containing active and inactive forms of the G protein Gαo

Dear Dr. Ajit,

We thank you for your email dated June 19th, 2025, and for the opportunity to revise and resubmit our manuscript entitled, “Rapid purification of brain protein complexes containing active and inactive forms of the G protein Gαo” (PONE-D-25-19360).

We are grateful to you and the reviewers for the insightful and constructive comments, which have helped us to significantly improve the clarity and quality of our manuscript. We have carefully considered all points raised and have revised the manuscript accordingly (with track changes). Please find below our detailed responses to each of the editor’s and reviewers’ comments.

Please note that editor’s and reviewers' comments are presented in blue font, while our responses are provided in standard black font. Corresponding changes about the reviewers' comments, along with the line numbers mentioned in the response, have been highlighted in red font in the main revised manuscript.

Please note that the corrected line numbers mentioned in our responses here correspond to the lines marked using the "Simple Markup" view under the "Review" tab in the revised manuscript. This helps maintain readability while still indicating where revisions have occurred. If this setting is not enabled, you can simply go to the Review tab, locate the Markup section, and in the first drop-down, Display for Review, select “Simple Markup” (for Microsoft Word 2021).

Response to #Editor’s comments:

Please ensure that your manuscript meets PLOS ONE's style requirements, including those for file naming. The PLOS ONE style templates can be found at https://journals.plos.org/plosone/s/file?id=wjVg/PLOSOne_formatting_sample_main_body.pdf and https://journals.plos.org/plosone/s/file?id=ba62/PLOSOne_formatting_sample_title_authors_affiliations.pdf

Response: We thank the editor for providing us with the templates. We have referred them and formatted the main text and affiliations according to the provided templates in the revised version of the manuscript.

2. To comply with PLOS ONE submissions requirements, in your Methods section, please provide additional information regarding the experiments involving animals and ensure you have included details on (a) methods of sacrifice, (b) methods of anaesthesia and/or analgesia, and (c) efforts to alleviate suffering.

Response: We thank the editor for highlighting this important requirement. We recognize the importance of providing clear and complete ethical information following PLOS ONE’s submission guidelines for animal research. We have added all the required information in the Materials and Methods Section.

Changes made in the manuscript: Added a sub-section in the Materials and Methods Section, “Animal Handling and Ethics Statement”. Added lines 126 to 135.

S.K.'s laboratory is funded by Department of Biotechnology (DBT), Ministry of Science and Technology India (BT/PR38584/MED/122/247/2020), Department of Science and Technology (DST), Ministry of Science and Technology India (CRG/2021/000732) and DBT/Wellcome Trust India Alliance (IA/I/22/2/506480). M.R.K. is supported by NIH grant R01NS086932.

Please provide an amended statement that declares all the funding or sources of support (whether external or internal to your organization) received during this study, as detailed online in our guide for authors at http://journals.plos.org/plosone/s/submit-now. Please also include the statement “There was no additional external funding received for this study.” in your updated Funding Statement. Please include your amended Funding Statement within your cover letter. We will change the online submission form on your behalf."

Response: We thank the editor for letting us know. We have provided the amended statement regarding the funding statement in the cover letter. Kindly include it in the manuscript from your end.

We thank Dr Prasad Abnave (NCCS, Pune) for critical evaluation and valuable feedback on the manuscript. We heartily thank our lab members, Ajay Pradhan, Vandan Maurya, and Debolina Sarkar, for their continuous and generous support, and Nikhat Khan for skillful technical assistance.

S.K.’s laboratory is funded by Department of Biotechnology (DBT), Ministry of Science and Technology India (BT/PR38584/MED/122/247/2020), Department of Science and Technology (DST), Ministry of Science and Technology India (CRG/2021/000732) and DBT/Wellcome Trust India Alliance (IA/I/22/2/506480). M.R.K. is supported by NIH grant R01NS086932.

S.K.’s laboratory is funded by Department of Biotechnology (DBT), Ministry of Science and Technology India (BT/PR38584/MED/122/247/2020), Department of Science and Technology (DST), Ministry of Science and Technology India (CRG/2021/000732) and DBT/Wellcome Trust India Alliance (IA/I/22/2/506480). M.R.K. is supported by NIH grant R01NS086932.

Response: We thank the editor for informing us. We have provided the amended statement regarding the Acknowledgements in the cover letter. Kindly include it in the manuscript from your end.

5. PLOS requires an ORCID iD for the corresponding author in Editorial Manager on papers submitted after December 6th, 2016. Please ensure that you have an ORCID iD and that it is validated in Editorial Manager. To do this, go to 'Update my Information’ (in the upper left-hand corner of the main menu), and click on the Fetch/Validate link next to the ORCID field. This will take you to the ORCID site and allow you to create a new iD or authenticate a pre-existing iD in Editorial Manager.

Response: We thank the editor for letting us know. We have validated our ORCID ID as per the given instructions.

6. Please include your full ethics statement in the ‘Methods’ section of your manuscript file. In your statement, please include the full name of the IRB or ethics committee that approved or waived your study, as well as whether or not you obtained informed written or verbal consent. If consent was waived for your study, please include this information in your statement as well.

Response: We thank the editor for bringing this point to our attention. We have included all the necessary information in the Methods Section.

Changes made in the manuscript: Added a sub-section in the Materials and Methods Section, “Animal Handling and Ethics Statement”. Added lines 126 to 135.

7. We notice that your supplementary table is included in the manuscript file. Please remove and upload with the file type ‘Supporting Information’. Please ensure that each Supporting Information file has a legend listed in the manuscript after the references list.

Response: We have removed the table from the manuscript and re-uploaded it with the file type “Supporting Information”. We have listed a legend for the supporting information after the References.

Changes made in the manuscript: Added lines 488 to 490 and uploaded supporting information

8. Please include captions for your Supporting Information files at the end of your manuscript, and update any in-text citations to match accordingly. Please see our Supporting Information guidelines for more information (http://journals.plos.org/plosone/s/supporting-information)

Response: We have made the necessary additions and provided in-text citations.

9. PLOS ONE now requires that authors provide the original uncropped and unadjusted images underlying all blot or gel results reported in a submission’s Figures or Supporting Information files. This policy and the journal’s other requirements for blot/gel reporting and figure preparation are described in detail at https://journals.plos.org/plosone/s/figures#loc-blot-and-gel-reporting-requirements and https://journals.plos.org/plosone/s/figures#loc-preparing-figures-from-image-files. When you submit your revised manuscript, please ensure that your figures adhere fully to these guidelines and provide the original underlying images for all blot or gel data reported in your submission. See the following link for instructions on providing the original image data: https://journals.plos.org/plosone/s/figures#loc-original-images-for-blots-and-gels. In your cover letter, please note whether your blot/gel image data are in Supporting Information or posted at a public data repository, provide the repository URL if relevant, and provide specific details as to which raw blot/gel images, if any, are not available. Email us at plosone@plos.org if you have any questions.

Response:

We thank the editor for letting us know. As per the journal’s requirement, we have uploaded all the raw blot and gel images as a separate file. We have also acknowledged the same in the cover letter.

Response to Reviewer #1 comments:

The manuscript authored by Yadav et al., entitled "Rapid purification of brain protein complexes containing active and inactive forms of the G protein Gαo" presents a study aimed at identifying downstream effectors of the Gαo protein, the most abundant neuronal heterotrimeric G protein. The authors use a novel biochemical approach to isolate native Gαo-GTP protein complexes from mouse brain lysates. Their findings rely on immunoprecipitation and size exclusion chromatography to analyze Gαo complexes, demonstrating enrichment in Gαo and depletion of other GTPYs. This study addresses a significant gap in understanding Gαo signaling, as no direct effectors have been well-characterized despite its abundance and major role in neuronal function. The methods are novel and well-executed, but several areas require clarification, additional controls, and improved discussion to strengthen the manuscript’s impact and rigor. My specific comments are as follows:

Response: We thank the reviewer for the critical evaluation and the constructive feedback provided for the manuscript. The suggestions have certainly helped us to increase the quality and clarity of the manuscript. Here, we provide a point-by-point response to each of the reviewer’s comments.

Please note that the corrected line numbers mentioned in our responses here correspond to the lines marked using the "Simple Markup" view under the "Review" tab in the revised manuscript. This helps maintain readability while still indicating where revisions have occurred. If this setting is not enabled, you can simply go to the Review tab, locate the Markup section, and in the first drop-down, Display for Review, select “Simple Markup” (for Microsoft Word 2021).

1. The introduction mentions genetic and interaction screens that failed to identify Gαo effectors, but does not compare the current method’s advantages over these approaches in detail. A stronger rationale for why biochemical purification will likely succeed where genetic screens failed would strengthen the manuscript. Expand the introduction to discuss the limitations of yeast two-hybrid and genetic screens more explicitly and highlight how the current method overcomes these.

Response: We thank the reviewer for this thoughtful and constructive comment. You're right that a more explicit comparison between classical genetic/interaction screens and our biochemical purification approach would enhance the introduction. As per your suggestion, we have also added information that explicitly explains studies done in yeast and C. elegans (Revised manuscript Lines 80 to 89). We have discussed the limitations of genetic screens (Revised manuscript Lines 94 to 97) and Y2H interaction systems in the revised version of the manuscript. (Revised manuscript, Lines 101-109). We have also added a paragraph that highlights how our method overcomes those limitations. (Revised manuscript, Lines 112 to 119).

Changes made in the manuscript: Added lines 80 to 89 (studies done in yeast and C. elegans) and references [17] and [18], with subsequent changes in reference numbers. Added lines 94 to 97 (Limitations of genetic screens), 101 to 109 (Limitations of Y2H systems), and 112 to 119 (Advantages of our method).

2. The manuscript aims to identify Gαo effectors but does not report any specific effector proteins identified through the described methods. The discussion mentions that effectors may be present at low abundance and detectable by mass spectrometry, but no such data are provided. This is a major limitation, as the study’s primary goal appears unfulfilled. If mass spectrometry was performed, include preliminary results or clarify why such data are not presented. If not performed, explicitly state this as a limitation and outline plans for future work to identify effectors using these methods.

Response: We thank the reviewer for raising this crucial point. Our goal was first to establish a reliable and reproducible protocol that could distinguish active Gαo-GTP complexes from the inactive Gαo-GDP state and thereby enable identification of direct Gαo effectors. Now that this method has been validated and optimized, we have validated a known effector of Gαo, GPRIN-1, performed mass spectrometry, and incorporated preliminary results into the revised version, as well as revised Fig. 1.

Changes made in the manuscript: Added lines 189 to 191 (Methods-information about anti-GPRIN-1 antibody), Lines 210-229 (Methods- Mass spectrometry protocol), Lines 283 to 301 (Results for GPRIN-1 western blot and mass spectrometry). Also added revised Fig.1 Panel E and G. Added lines 463 to 465 (figure legend- Revised Fig. 1, Panel E), and lines 466 to 470 (figure legend- Revised Fig. 1, Panel G).

3. The observed molecular weights of Gαo complexes (1249–945 kDa for GDP-bound and 98–398 kDa for GTPγS-bound) are much larger than expected, even accounting for Triton X-100 micelles. The manuscript attributes this to potential effector interactions but does not adequately discuss alternative explanations, such as non-specific protein aggregation or micelle size variability. Include controls to rule out non-specific interactions, such as comparing Gαo complexes to those of other G proteins (e.g., Gaq or Gai) under identical conditions. Discuss the impact of detergent micelles on apparent molecular weight more thoroughly, potentially referencing studies on membrane protein complexes in similar systems.

Response: We thank the reviewer for bringing this point to our attention. The apparent molecular weights of Gαo protein complexes observed in our size exclusion chromatography (SEC) experiments are markedly higher than the expected sizes for the Gαo monomer or Gαo-βγ heterotrimer. While we interpret this shift as binding of unknown effectors upon Gαo activation, we have also discussed that these elevated values likely also reflect the contribution of detergent micelles, which are known to influence the hydrodynamic behaviour of solubilized membrane proteins. As per your suggestion, we have also added alternative explanations that could be potentially responsible for the change in molecular weights of the complexes.

Changes made in the manuscript: Added lines 323 to 326.

4. Validate anti-Gαo antibody specificity by testing it against lysates from Gao knockout mice or performing co-immunoprecipitation with other G protein subunits (e.g., Gaq or Gai) to confirm selectivity.

Response: We thank the reviewer for this important comment. We agree that validating the specificity of the anti-Gαo antibody is critical to ensuring the reliability of our

---

## [Decision Letter · Decision Letter 1]

17 Nov 2025

Rapid purification of brain protein complexes containing active and inactive forms of the G protein Gαo

PONE-D-25-19360R1

Dear Dr. Kumar,

We’re pleased to inform you that your manuscript has been judged scientifically suitable for publication and will be formally accepted for publication once it meets all outstanding technical requirements.

Kind regards,

Ajit Prakash, PhD

Academic Editor

PLOS ONE

Additional Editor Comments (optional):

Reviewers' comments:

Reviewer's Responses to Questions

**Comments to the Author**

Reviewer #2: (No Response)

Reviewer #3: All comments have been addressed

2. Is the manuscript technically sound, and do the data support the conclusions?

Reviewer #2: Yes

Reviewer #3: Yes

3. Has the statistical analysis been performed appropriately and rigorously?

Reviewer #2: Yes

Reviewer #3: Yes

4. Have the authors made all data underlying the findings in their manuscript fully available?

Reviewer #2: Yes

Reviewer #3: Yes

5. Is the manuscript presented in an intelligible fashion and written in standard English?

Reviewer #2: Yes

Reviewer #3: Yes

Reviewer #2: N/A

Reviewer #3: (No Response)

**Do you want your identity to be public for this peer review?** For information about this choice, including consent withdrawal, please see our Privacy Policy

Reviewer #2: No

Reviewer #3: No

---

## [Editor Report · Acceptance letter]

PONE-D-25-19360R1

PLOS ONE

Dear Dr. Kumar,

I'm pleased to inform you that your manuscript has been deemed suitable for publication in PLOS ONE. Congratulations! Your manuscript is now being handed over to our production team.

Kind regards,

on behalf of

Dr. Ajit Prakash

Academic Editor

PLOS ONE